# Rationally engineered active sites for efficient and durable hydrogen generation

Yurui Xue [1], Lan Hui[1], Huidi Yu[1], Yuxin Liu[1], Yan Fang[1], Bolong Huang [2], Yingjie Zhao[3], Zhibo Li[3] & Yuliang Li [1,4]

The atomic-level understanding of the electrocatalytic activity is pivotal for developing new metal-free carbon electrocatalysts towards efficient renewable energy conversion. Here, by utilizing the amidated-carbon fibers, we demonstrate a rational surface modulation strategy on both structural and electronic properties, which will significantly boost the hydrogen evolution reaction activity of electrocatalysts. Theoretical calculations reveal the amidation decorated surface will promote significantly more 2D electrons towards the localization at the $C=O$ branch. The modified surface displays a self-activated electron-extraction characteristic that was actualized by a fast reversible bond-switching between $HO-C=C_{catalyst}$ and $O=C-C_{catalyst}$. Experimentally, this metal-free electrode exhibits outstanding hydrogen evolution reaction activities and long-term stabilities in both acidic and alkaline media, even surpassing the commercial 20 wt% Pt/C catalyst. Thus, this strategy can extend to a general blueprint for achieving precise tuning on highly efficient electron-transfer of hydrogen evolution reaction for broad applications under universal pH conditions.

[1] Institute of Chemistry, Chinese Academy of Sciences, 100190 Beijing, PR China. [2] Department of Applied Biology and Chemical Technology, The Hong Kong Polytechnic University, Hung Hom, Kowloon 99907, Hong Kong. [3] School of Polymer Science and Engineering, Qingdao University of Science and Technology, 266042 Qingdao, P.R. China. [4] University of Chinese Academy of Sciences, 100049 Beijing, PR China. Correspondence and requests for materials should be addressed to Y.X. (email: xueyurui@iccas.ac.cn) or to B.H. (email: bhuang@polyu.edu.hk) or to Y.L. (email: ylli@iccas.ac.cn)

Electrochemical water splitting ($2H_2O \rightarrow 2H_2 + O_2$) is a desirable and facile method to produce $H_2$ (hydrogen evolution reaction, HER), which possesses great attraction for energy storage due to the high energy density (120–142 MJ $kg^{-1}$)[1–5]. In this process, an electrocatalyst with lower overpotential is of particular importance[6–11]. Extensive efforts have been devoted to the exploration of precious-/nonprecious-metal electrocatalysts, however, most of them must be immobilized on conductive supports (e.g., glassy carbon electrode, Cu foam, Ni foam, etc.)[12–16]. The challenge of inevitable peeling of the active components from the supports will substantially reduce the activity and lifetime the electrocatalysts. Besides, the high cost and scarcity of the noble-metal electrocatalysts severely restrict their cosmically commercial applications. Thus, developing metal-free, self-supported electrocatalysts with high activity and stability has been expected to be an ideal method to resolve these problems[2,3,17–20].

The three-dimensional carbon fiber cloth (CFC), a metal-free flexible network of intertwined nanofibers comprising of sheets of carbon atoms arranged in the regular hexagonal pattern (graphene sheets), exhibits inspiring properties including high conductivity, high corrosion resistance, and high chemical and mechanical stability, which all imply that CFC has the potentials to be an efficient HER electrocatalyst. Unfortunately, CFC is commonly used as a substrate for supporting electrocatalysts during the past decades. Up to now, however, there is still no report of direct utilization of metal-free CFC as HER cathode.

In electrocatalysis, HER proceeds through the first formation of adsorbed hydrogen intermediates ($H_{ad}$, Volmer step) on catalyst surface, followed by either the rate-determining Tafel step ($H_{ad} + H_{ad} \rightarrow H_2$) or the Heyrovsky step ($H^+ + H_{ad} + e^- \rightarrow H_2$ in acidic electrolyte or $H_2O + H_{ad} + e^- \rightarrow H_2 + OH^-$ in alkaline electrolyte)[8,9]. All above-mentioned relies on the optimal electron transfer behavior of the electrocatalyst. The catalyst surface, where the charge transfers, reactant adsorption and product desorption occur, lies at the core of the electrochemistry[8,21]. The chemical functionalization provides the rational modulation on surface characteristics such as the improvement of electrons transfer, the increasing of active sites and the accessibility of the electrolyte to its surface, thus improving the electrocatalytic performance[8,22–25]. More importantly, by applying the chemical functionalization strategy, we can prepare an electrocatalyst with determined active sites structure, which is critical for reinforcing the understanding of the catalytic mechanisms at the atomic level. Meanwhile, achieving a wide range of pH condition applications of HER electrode development would definitely accelerate the excavation of future low-temperature solid oxide fuel cell excavation[26].

Herein, we employ amide group functionalized CFC (A-CFC) to access the HER catalytic activity and investigate the origin of superior electrocatalysis from the atomic level. The synthesized A-CFC electrocatalyst displays a well-defined structure with definite active sites. DFT calculations reveal that the introduction of amide groups evidently optimize electronic properties on the surface with enhanced delocalization of 2D electrons. The fast electron-transfer controlled by the reversible resonance bond switch lays a good foundation for the exceptional HER performance under all pH conditions. The high HER electroactivity of A-CFC in acidic and alkaline conditions have been verified by the small overpotentials of 78 and 71 mV at 10 mA cm$^{-2}$ in acidic and alkaline conditions, respectively, which are superior to those reported metal-free electrocatalysts as well as most metal-based ones. Even at higher overpotentials, A-CFC presents much better HER activities than that the commercial 20 wt% Pt/C.

## Results

**Electronic activities and adsorption energetic trends for HER.** We reasoned the high HER performance from DFT calculations (Supplementary Note 1). The geometry optimized local structure for the group (–$CONH_2$) on the catalyst surface has been illustrated. With comparing the binding energy, we find the $H^+$ can be favorably located on the O site from the C=O of the group (Fig. 1a). We point out that the dominant charge-density distribution on the group (–$CONH_2$) does not match the HOMO level and are deeper away from the Fermi level ($E_F$). This guarantees the electron-rich character, but also provides a better s-p charge transfer for proton–electron exchange. Meanwhile, the HOMO and LUMO orbitals near the $E_F$ are complementary to each other. This implies that the electronic orbital charge distributions near the $E_F$ have been totally modified with asymmetrically distribution character exhibited (Fig. 1b).

The projected partial density of states (PDOSs) compares individual contributions of s-orbitals and p-orbitals from C, N, and O, respectively. The O-2p band differs from the N-2p band possessing a distinguished peak at $E_V$ −6.0 eV ($E_V = 0$ eV for $E_F$). The p-state centering at $E_V$ −2.0 eV is nearly the same for both N and O denoting a common character of lone-pair p-electrons. The N-2p bonding state ($E_V$ −9.3 eV) stays lower than the O-2p indicating the over-stronger binding between proton and (–$NH_2$) than the –C=O branch. Regarding the C-site from the group, the bonding state stays much lowered and broadened at $E_V$ −6.0 eV. For better reference of s–p orbital overlapping behavior, we take the H near the O=C and $NH_2$ branches to compare H-1s electronic contributions. We found the H tends to form s–p overlapped scenarios for both cases. The overlapped N–$H_{ad}$ and O–$H_{ad}$ are staying at $E_V$ −10.0 eV and $E_V$ −6.0 eV, respectively. This confirms that the H-adsorption on the group is chemisorption with substantial s–p orbital overlapping, where the charge transfer occurs between either H and N or H and O. However, further formation energy analysis comparison demonstrates the O–H binding is energetically more favorable and will not form an over-binding effect at the case of high acidic H-coverage. We also experimentally observed amount of protonated amides after the acidic HER owing to the over-binding effect (Fig. 1c). For the electron-transfer during HER, we compared three simple steps as: before-adsorption, H-adsorption, and 2H-adsorption, at the region when two groups are attached on the CFC surface. Considering the illustration of PDOS of H-s and O-p states, we find the charge transfer can be fulfilled directly from the O-2p non-bonding long-pair electronic orbital component ($E_V$ −2.0 eV) onto the spσ state ($E_V$ −6.0 eV). This deduction arises because of a lowered p-π lone-pair state has been shown at the H-adsorption step. After the 2H-adsorption step, the long-pair state recovers while the spσ bonding state intensifies from the orbital levels of $E_V$ −4.0 eV to $E_V$ −9.0 eV showing evident chemical adsorptions (Fig. 1d).

We move onto the energetics on HER performance. The formation energies of H on the CFC with different ways of surface modifications have been preliminarily benchmarked. These adsorption cases are classified with five different models. Here we mainly discuss the chemisorption difference and variation behaviors relative to the thermoneutral line ($\Delta G = 0$ eV)[5]. Firstly, the N-doped (N) system shows positive adsorption energy from 0.48 to 0.62 eV. For the tri-N-doped (3N), it is the system where three N substitutes the 2-fold coordinated C-sites neighboring to the C-vacancy ($V_C$) showing the overbinding adsorption strength ranges from −1.72 to −2.09 eV. For the (2C+N) system, it is the CFC with C-vacancy ($V_C$) where one of 2-fold coordinated C-sites has been replaced by N. The adsorption energy is from −0.24 to −0.50 eV. The (3C) system is the CFC

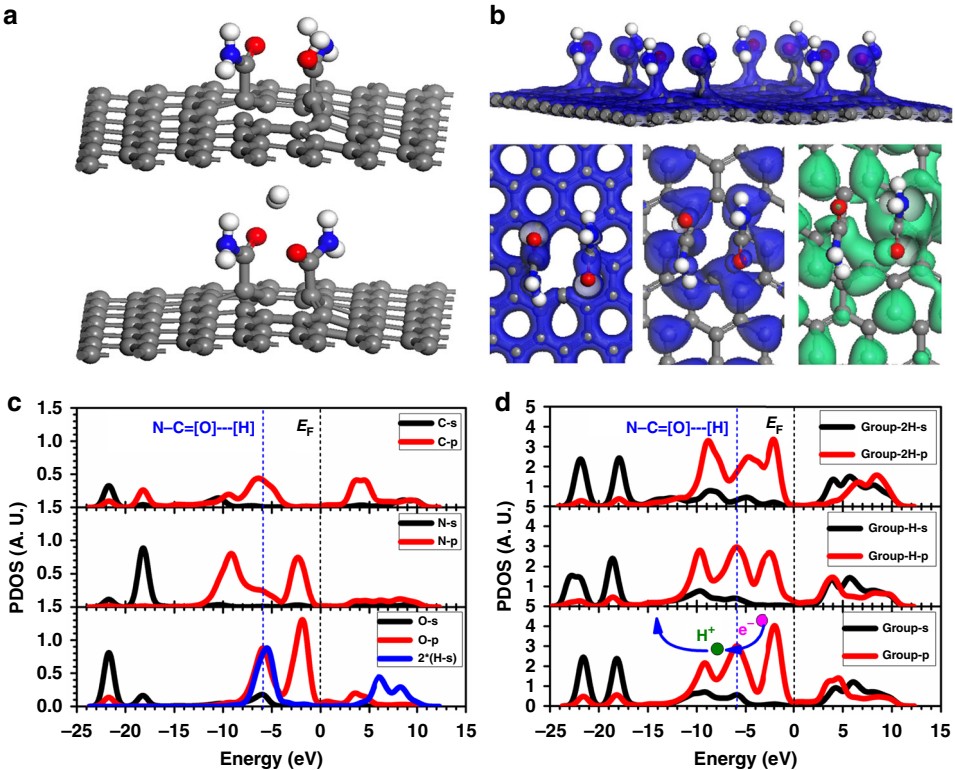

**Fig. 1** Electronic structures and projected density of states for HER process. **a** Geometry optimized paired groups (CONH$_2$) with mono-H adsorption and H$_2$ stabilized between them. **b** The orbital charge densities for efficient s-p band overlapping for proton–electron exchange. **c** PDOSs comparison of the group. **d** PDOS variations of the paired groups during the HER process

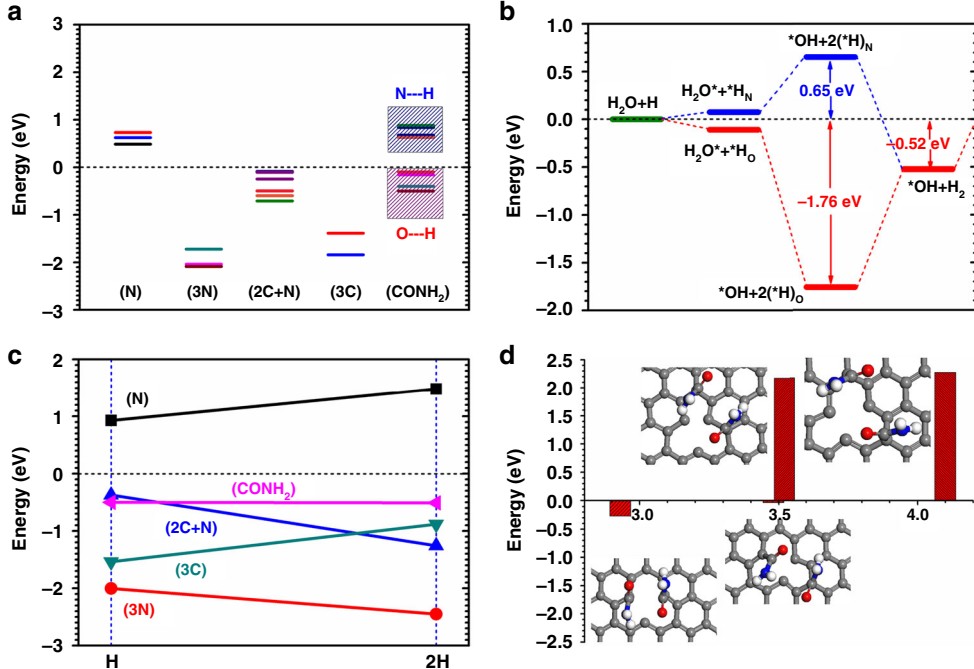

**Fig. 2** Adsorption energetic trend and pathways on amidation modified catalyst. **a** Formation energies benchmarks of H-adsorption on five different model systems. **b** The group (CONH$_2$) catalyzed HER formation energy pathway under the alkaline condition. **c** The chemisorption energy slope comparisons among five different model systems. **d** The potential H-bond influence on a pair of functional groups structural variation

with C-vacancy ($V_C$) where three neighboring C-sites are 2-fold coordinated. It shows the adsorption energy from −1.39 to −1.84 eV, which is also an overbinding system. The (CONH$_2$) system is the CFC attaching with the group showing two different classes of H-adsorption energies. For the O–H binding case, the energy is ranging from −0.10 to −0.50 eV while the N–H case shows the range from 0.63 to 0.88 eV. Therefore, the C=O branch is energetically favorable for H-adsorption (Fig. 2a).

The energetic pathway for HER within the alkaline condition has been illustrated. It is overall energetically favorable when the HER performs via the C=O branch of the group. The process of $H_2O$ splitting contributes a relatively large energy gain ($-1.76$ eV), while this step confronts a barrier of 0.65 eV for the HER performing via $NH_2$ branch. The overall reaction is exothermic with $-0.52$ eV gained in energy (Fig. 2b). Considering the acidic HER performance, the chemisorption energy comparison is the determinant factor. We connect the adsorption energies with easily comparing the slopes, which demoting the abilities for efficient proton–electron exchange and desorption. The positive slope denotes the trend for desorption, while negative one tends to be strongly bonded with catalyst support. With this trend, the (3N) system is an overbinding system with even stronger adsorptions at high H-coverage. The (N) system barely provides any effective adsorption for H. Negative slope for the (2C+N) system shows a trend that it only prefers low H-coverage before overbinding at high H-coverage. Considering the (3C) system, positive slope demonstrates that the HER will be improved at high H-coverage. The slope for the group ($CONH_2$) system shows almost unchanged and nearly parallel to the thermoneutral line ($\Delta G = 0$ eV). This indicates the system not only provide an optimal adsorption strength but also contribute efficient desorption with chemisorption energy close to the $\Delta G = 0$ eV (Fig. 2c), following similar discussion from Nøskov et al.[5] Further analysis on the energy trend with related to the structural difference, we find out some preliminary information on the weak H-bond impact between adjacent groups on the CFC. The O from the C=O branch will form a weak inter-molecular H-bond to further stabilize the H from the $NH_2$ branch from a nearby group. The equilibrium inter-molecular distance is estimated to be ~2.9 Å. The analysis of the H-bond influence is to further demonstrate the possible charge-transfer between different neighboring groups from the CFC. This is significant to guarantee a substantial electron-rich center for actively charge-transfer during the HER process (Fig. 2d).

**Synthesis and structural characterization.** Inspired by these calculation results, we prepared amide-functionalized carbon fiber cloth (A-CFC) through a simple carboxylation-amidation strategy (Fig. 3a, b; see Methods for details) to verify these results. Optical photographs, scanning electron microscopy (SEM) and transmission electron microscopy (TEM) were employed to characterize the morphologies of the as-synthesized samples. Figure 3a shows the photographs of pristine CFC, exhibiting its flexibility and porous nature, which have been well maintained even after carboxylation (Fig. 3c, d and Supplementary Fig. 1) and amidation treatments (Fig. 3f, g and Supplementary Fig. 1). It can be seen from the SEM images that pure carboxylated CFC (c-CFC, Fig. 3c, d) has a smooth surface with shallow grooves, while the A-CFC possesses a rougher, bumpy surface with deepened grooves (Fig. 3f, g), which indicate the formation of more defective sites after amidation and thus the increase of the number of active sites. In Raman spectra, both c-CFC and A-CFC exhibit three prominent peaks at 1365.5 (D peak), 1594.3 (G peak) and 2727.3 (2D peak), respectively (Supplementary Fig. 2). The larger intensity ratio of D-peaks and G-peaks ($I_D/I_G$) for A-CFC (0.97) than c-CFC (0.88) confirms the higher defect density of the A-CFC[27]. Interestingly, pristine c-CFC is hydrophobic with a contact angle (CA) of $125.1 \pm 1.5°$ (Fig. 3e); after amidation, the obtained A-CFC becomes complete wetting with a CA of 0° (Fig. 3h and Supplementary Fig. 3), which should be very helpful for the accessibility of electrolyte to active sites on A-CFC and the mass/ion diffusion. These observations are beneficial for enhancing the overall electrocatalytic performances of the electrocatalyst.

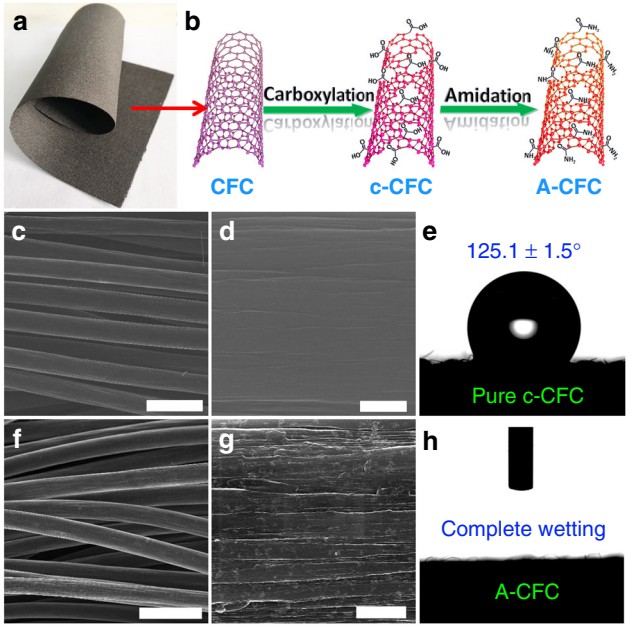

**Fig. 3** SEM and surface wettability characterization of the samples. **a** Optical image of pristine CFC. **b** The two-step strategy for the synthesis of A-CFC, low-magnification, and high-magnification SEM images of **c, d** c-CFC and **f, g** A-CFC. **e** Pristine c-CFC is hydrophobic with a CA of $125.1 \pm 1.5°$. **h** After amidation, the obtained A-CFC becomes complete wetting with a CA of 0°. Scale bars: **c, f** 30 μm; **d, g** 1 μm

High-resolution TEM images (Fig. 4a, b) further corroborated the defect-rich edges of A-CFC and showed that the A-CFC had curved streaks with the spacing distance of 3.39 Å (Fig. 4b). Scanning TEM image and energy dispersive spectroscopy (EDS) mapping analysis (Fig. 4c) revealed the N elements uniformly distributed throughout the nanofibers. X-ray photoelectron spectroscopy (XPS) survey of A-CFC shows the presence of C, N, and O elements (Fig. 4d), while the XPS survey of c-CFC has only C and O elements. This solidly confirmed that there are no any metal impurities in the materials used in our work. In addition, the high-resolution N 1s spectrum showed two peaks at 399.5 and 401.7 eV corresponding to neutral amine nitrogen and ammonium ions[28], verifying that the nitrogen is in the amide form (Fig. 4e). This was also verified by the O 1s spectra in Supplementary Fig. 4. C1s peak can be fitted into five peaks at 284.4, 285.2, 286.9, 288.8, and 290.9 eV, respectively, corresponding to aliphatic carbon chains (C–C,H), amine groups (C–N), carbonyl (O=C), carboxyl carbon (O=C–N), and the π–π* transitions in amides (Fig. 4f)[28]. These observations demonstrated the successful anchoring of amide groups on carbon nanofiber cloth (A-CFC).

**HER electrocatalytic performances.** The HER performance of A-CFC as a three-dimensional electrode was investigated in $H_2$-saturated acidic condition (0.5 M $H_2SO_4$) using a typical three-electrode system (Supplementary Fig. 5). The freshly-prepared A-CFC was directly used as a cathodic electrode and the san rate is set to 1 mV s$^{-1}$ to minimize the capacitive current. The 20 wt% Pt/C was used for comparison. Pristine CFC (Supplementary Fig. 6a) and pure c-CFC (Fig. 5a) show negligible HER activities with large overpotentials of 577 and 457 mV at the current density of 10 mA cm$^{-2}$ ($j_{10\,mA\,cm^{-2}}$), respectively. As expected, amide-functionalized CFC (A-CFC) shows a significantly enhanced HER activity with a lower overpotential of 78 mV at 10 mA cm$^{-2}$. Remarkably, the HER

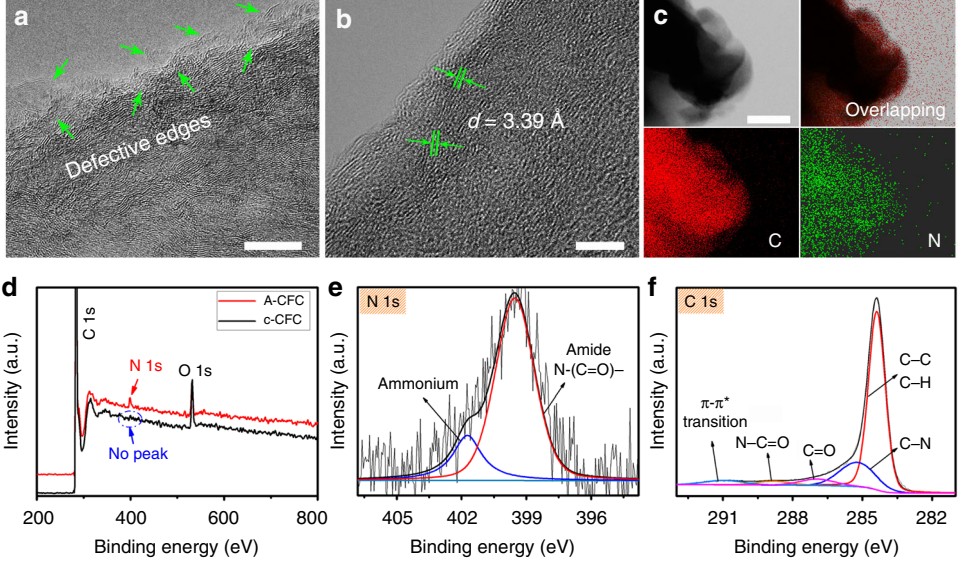

**Fig. 4** TEM and XPS characterization of the A-CFC cathode. **a**, **b** High-resolution TEM images of A-CFC. **c** Scanning TEM image and corresponding elemental mapping of N (yellow) and C (green) elements in A-CFC. Scale bars: **a** 10 nm; **b** 5 nm; **c** 200 nm. **d** XPS survey spectra of c-CFC (dark line) and A-CFC (red line). High resolution **e** N 1s and **f** C 1s XPS spectra of A-CFC

activity of A-CFC even surpasses that of 20 wt% Pt/C as the applied potentials over −0.28 V (vs. RHE). Besides, the A-CFC shows a small Tafel slope of 37 mV dec$^{-1}$ (Fig. 5b), which is very close to that of Pt/C (30.7 mV dec$^{-1}$), where the Heyrovsky and Tafel reactions are the rate-determining step[29]. By extrapolating the Tafel plots to the overpotential of zero (Supplementary Fig. 6b), the exchange current density ($j_0$) of A-CFC was calculated to be 0.1 mA cm$^{-2}$ which is about 278 times higher than c-CFC ($3.6 \times 10^{-4}$ mA cm$^{-2}$), suggesting more favorable HER kinetics at the A-CFC/electrolyte interface. These values are smaller than the reported metal-free HER electrocatalysts including 3D graphene networks[27] and C$_3$N$_4$@NG hybrid[20], and most of metal-based ones such as Pt NWs/SL-Ni(OH)$_2$[30], CoP/NCNHP[13], CoPS[31], Ni$_{4.5}$Fe$_{4.5}$S$_8$[32], Co-NG[33], MoC$_x$ nano-octahedra[34], and CoN$_x$/C[35] (Supplementary Table 1). The smaller Tafel slope (Fig. 5b) and larger $j_0$ (Supplementary Fig. 6; Table 1) of A-CFC confirm its higher catalytic efficiency and more favorable reaction kinetics in the HER process. The stability of A-CFC was further examined (Fig. 5c). There is no change in current density even after 13,000 continuous cycles. Durability measurement was then conducted on the A-CFC (inset in Fig. 5c). It can be seen that the current density decreased at the beginning which might be due to the H$_2$ adsorption on the electrode surface and then stabilized over 28 h. SEM images (Supplementary Fig. 7), elemental mapping analysis (Supplementary Fig. 8), and XPS results (Supplementary Fig. 9) showed that there are no any morphological and chemical changes of the A-CFC.

The electrocatalytic HER performance of A-CFC was further investigated in H$_2$-saturated alkaline electrolyte (1.0 M KOH). Pristine CFC (Supplementary Fig. 10a) and c-CFC (Fig. 5d) require large overpotentials of 611 mV and 530 mV to achieve $j_{10\,mA\,cm^{-2}}$, respectively. Remarkably, A-CFC shows a high HER activity with an extreme small overpotential of 71 mV at $j_{10\,mA\,cm^{-2}}$, which is smaller than 20 wt% Pt/C (79 mV) and other benchmarked metal/metal-free electrocatalysts in alkaline conditions including NiFe-LDH/NF (210 mV)[1], Fe-Ni@NC-CNTs (202 mV)[36], and δ-FeOOH NSs/NF (108 mV).[37] Besides, the smaller Tafel slope value of A-CFC (54.5 mV dec$^{-1}$, close to that of Pt/C, Fig. 5e) and the larger $j_0$ (0.73 mA cm$^{-2}$, Supplementary Fig. 10) than pure c-CFC

reveals the favorable HER kinetic process in alkaline media. We further access the long-term stability of A-CFC in 1.0 M KOH (Fig. 5f). After 18,000 continuous cycles, there is not any decreasing in electrocatalytic activity can be observed. Also, a durability test was conducted on A-CFC at applied potential of −1.1 V (vs. SCE) for more than 9 h (inset in Fig. 5f). A-CFC exhibits almost no variation in current density, suggesting its excellent stability in alkaline condition. After the cycling test, the XPS results (Supplementary Fig. 11) reveal that the C 1s, N 1s, and O 1s have almost no changes, indicating structural stability of the sample. The elemental mapping results further confirmed the XPS results (Supplementary Fig. 12). Additionally, the catalytic performance of the electrocatalysts have been determined in neutral conditions (1 M PBS, pH 7, Supplementary Fig. 13). A-CFC exhibits small overpotentials of 230.8 and 436 mV at the current densities of 2 and 10 mA cm$^{-2}$, respectively, and Tafel slope of 263.8 mV dec$^{-1}$. These values are better than that of c-CFC (570 and 791 mV at 2 and 4 mA cm$^{-2}$, respectively; Tafel slope: 730.4 mV dec$^{-1}$), and superior to that of non-metal and transition metal based electrocatalysts (Supplementary Table 2).

To better understand the origin of the intrinsic HER activity of A-CFC, electrochemical impedance spectroscopy (EIS) was conducted and fitted by the two-time constant parallel model (2TP, Fig. 5g, h; Supplementary Table 3) containing the solution resistance ($R_s$), the charge transfer resistance ($R_{ct}$), and the hydrogen adsorption resistance ($R_h$). The A-CFC shows much smaller $R_s$ (3.14 Ω) and $R_{ct}$ (0.69 Ω) than that of pure c-CFC ($R_s$ = 5.39 Ω; $R_{ct}$ = 0.81 Ω), suggesting the facilitated HER kinetics. The much smaller $R_h$ of A-CFC reveals the more efficient for the hydrogen intermediates adsorption on the catalyst surface, which is beneficial to the overall HER performances. The electrochemical surface area (ECSA) was estimated by measuring the double-layer capacitance ($C_{dl}$) through the CV method (Fig. 5i and Supplementary Fig. 14). As shown in Fig. 5i, the $C_{dl}$ of A-CFC is 9.4 mF cm$^{-2}$ which is about 4.3 times than that of pure c-CFC (2.2 mF cm$^{-2}$). As a result, the ECSAs of A-CFC and c-CFC were calculated to be 235 and 55 cm$^2$. The results suggest that the main origin of the remarkable improvement of the HER catalytic activity on A-CFC. In other words, the amidation can greatly increase number of the electrocatalytically active sites for efficient

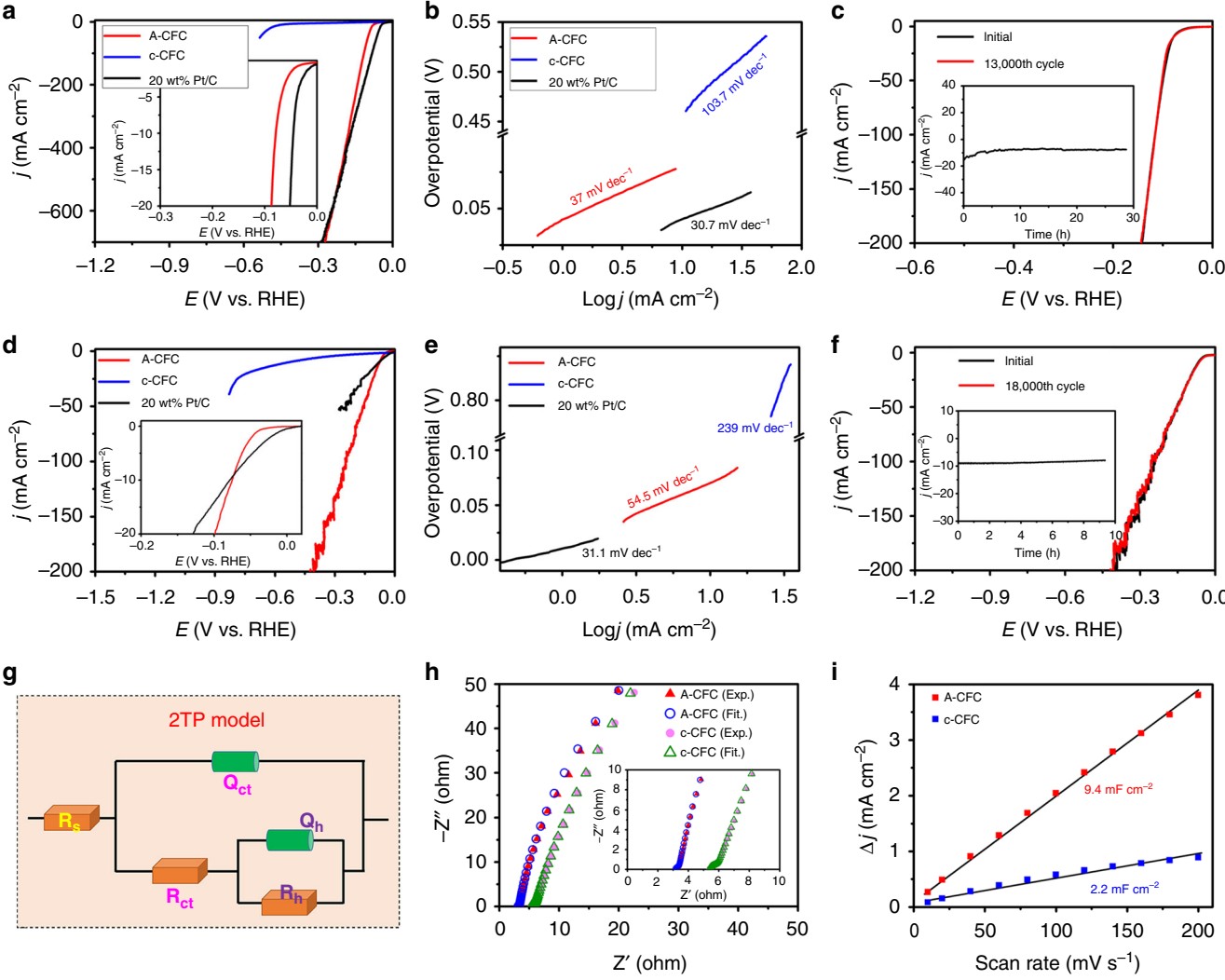

**Fig. 5** HER performance of A-CFC. HER polarization curves of catalysts obtained in **a** 0.5 M $H_2SO_4$ and **d** 1.0 M KOH; **b**, **e** corresponding Tafel plots of **a**, **d**; **c** The polarization curves of A-CFC obtained before and after 13,000 cycling test in 0.5 M $H_2SO_4$ (inset: current density-time curve of A-CFC in 0.5 M $H_2SO_4$); **f** The polarization curves of A-CFC obtained before and after 18,000 cycling test in 1.0 M KOH (inset: current density-time curve of A-CFC in 1.0 M KOH); **g** The equivalent circuit model; **h** Electrochemical impedance plots and **i** double layer capacitance for the as-synthesized samples (inset in **h**: enlargement of the high frequency region)

hydrogen generation. The A-CFC was determined to evolve hydrogen with high Faradaic efficiencies over 97% in both 0.5 M $H_2SO_4$ and 1 M KOH electrolytes (Supplementary Fig. 15). The excellent HER activity of A-CFC is also evidenced by its large specific current density (Supplementary Table 4). Mass activity is another critical criterion to evaluate the catalytic performance of a catalyst in practical uses[38,39]. Normalized to mass loading (Supplementary Fig. 16), A-CFC exhibited higher mass activities toward HER than most benchmarked metal-free electrocatalysts in both acidic and alkaline conditions (Supplementary Table 5). To further assess the intrinsic HER kinetics, we calculated the turnover frequencies (TOFs) of the A-CFC in different electrolytes (Supplementary Note 2). The TOF of the A-CFC catalyst is 0.86 and 0.41 $s^{-1}$ at the overpotential of 100 mV in 0.5 M $H_2SO_4$ and 1.0 M KOH, respectively, which are more than 10-fold and 2-fold larger than the c-CFC (0.08 $s^{-1}$ in 0.5 M $H_2SO_4$ and 0.14 $s^{-1}$ in 1.0 M KOH). This significant increase of TOF reveals an improved energetics at amidated interface during the hydrogen-evolving process, leading to the enhancement of HER catalytic performance of the A-CFC sample.

In summary, the A-CFC as an efficient metal-free electrocatalyst for HER in both alkaline and acidic media has been demonstrated. The superior HER activity has been attributed to the electron-extraction from by the self-activated bond-switching effect that originated from the amide modification. The surface 2D electron distribution is optimized that becomes delocalized towards the functional groups to reactivate the graphene systems. Thus, this general surface modification method will offer an electron-transfer highway via the p–p resonance switch robustly boost-up the HER performance within a wide range of pH conditions. More importantly, we have clarified the role and the mechanism of amide in efficient HER process, which provides valuable guidelines for the design and synthesis of efficient electrocatalysts.

## Methods

**Materials**. Reagents and solvents were analytical grade and used as received from the commercial suppliers. Carbon fiber cloth (CFC) was purchased from CeTech Co., Ltd. Deionized water (DI-$H_2O$) produced from a Millipore Milli-Q water purification system was used to prepare electrolytes. The calculation details are given in the Supplementary Materials.

**Preparation of amide-functionalized CFC (A-CFC).** The A-CFC was synthesized through a simple carboxylation-amidation strategy. In a typical procedure, a piece of carbon fiber cloth (5 cm × 5 cm, CFC) was subsequently cleaned with ethanol, Millipore water and ethanol under ultrasonic, and dried in an oven before use. The freshly cleaned CFC was then added into a stainless steel autoclave (50 mL) containing $HNO_3$ (40 mL), and was hold at 120 °C for 20 h. The resulted c-CFC was then washed thoroughly with Millipore water.

The freshly-prepared c-CFC was immediately into another Autoclave (50 mL) containing 40 mL ammonium hydroxide solution (28% $NH_3$ in $H_2O$). After 20 min standing, the sealed Autoclave was placed at 150 °C for 10 h, then was naturally cooled to room temperature and washed thoroughly with Millipore water, followed by being dried in a vacuum oven for 2 h. The A-CFC was obtained.

**Sample characterizations.** Scanning electron microscopy (SEM) images were recorded using a field emission scanning electron microscope (FESEM, Hitachi S-4800). Transmission electron microscopy (TEM) and high-resolution TEM (HRTEM) images were taken on a JEOL-2100F microscope. X-ray photoelectron spectroscopy (XPS, Kratos Axis Ultra DLD) was employed to determine the chemical composition and element states. The powder X-ray diffraction (XRD) experiments were carried out with a high resolution X-ray diffraction system using Cu Kα radiation ($\lambda = 0.15406$ nm). Raman spectra were recorded on a Renishaw-2000 Raman spectrometer using the 514.5 nm line of an Argon ion laser.

**Electrochemical measurements.** Electrochemical measurements were conducted on a typical three-electrode system (CHI 760E, CH Instruments) with a working electrode, a carbon rod counter electrode, and a saturated calomel electrode (SCE) reference electrode. The as-prepared samples were directly used as the working electrode. Linear sweep voltammetry (LSV) measurements were carried out in electrolyte (0.5 M $H_2SO_4$ or 1.0 M KOH) at a scan rate of 1.0 mV s$^{-1}$. All electrochemical tests were carried out in the $H_2$-saturated electrolyte. The average mass loading of A-CFC is 1.25 mg cm$^{-2}$. The A-CFC with 0.1 cm$^{-2}$ (geometric area) was used as the working electrode and the HER performances were investigated using a typical three-electrode system. For comparison, c-CFC with the similar mass loading (1.30 mg cm$^{-2}$) and the same geometric (0.1 cm$^{-2}$) was used the cathodic electrode. Current density normalized to geometric area of the working electrode. All LSV curves with 100% $iR$ compensation were obtained. Without further specified, all potentials were relative to the reversible hydrogen electrode (vs. RHE): $E_{RHE} = E_{measured} - iR + E_{SCE} + 0.059 * $ pH, where $E_{measured}$ is the measured potential, $i$ is the current, and $R$ is the ohmic drop tested by electrochemical impedance spectroscopy (EIS). The EIS data were obtained in the same configuration at the applied potential of $-0.1$ V (vs. RHE) in the frequency range from 100 kHz to 1 Hz with a signal amplitude perturbation of 5 mV.

**Calculation of electrochemically active surface area (ECSA).** The ECSA was measured by CV at no apparent Faradaic potential range of 0.5–0.6 V with different scan rates of 10, 20, 40, 60, 80, 100, 120, 140, 160, 180, and 200 mV s$^{-1}$. By plotting the current density $\Delta j$ at 0.55 V against the scan rates, the double layer capacitance ($C_{dl}$) was obtained. The ECSA was calculated by dividing $C_{dl}$ by the specific capacitance value. The specific capacitance for a flat surface is normally taken to be in the range of 20–60 μF cm$^{-2}$. In this study, we assume 0.040 mF cm$^{-2}$ for the calculation of ECSA. The ECSA for A-CFC and c-CFC were calculated to be 235 and 55 cm$^2$, respectively.

## Data availability

The data that support the plots within this paper and other finding of this study are available from the corresponding author upon reasonable request.

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

## Acknowledgements

This work was supported by the National Nature Science Foundation of China (21790050 and 21790051), the National Key Research and Development Project of China (2016YFA0200104), and the Key Program of the Chinese Academy of Sciences (QYZDY-SSW-SLH015).

## Author contributions

Y.Li conceived and designed the research and analysis, reviewed and edited this manuscript. Y.X. conducted the catalyst synthesis, electrochemical experiments, analyzed the data, and wrote the draft. B.H. carried out the theoretical calculations, analyzed the data, and wrote corresponding discussions. L.H., H.Y., Y.Liu, Y.F., Y.Z., and Z.L. gave useful help during the experiments.

## Additional information

**Competing interests:** The authors declare no competing interests.

