## [Peer Review File · Nature Communications]

Reviewers' comments:

Reviewer #1 (Remarks to the Author):

The authors suggest amidated-carbon fibers as a metal-free electrocatalyst for hydrogen evolution reaction (HER). Impetus to synthesizing this material is based on the reported computational study showing favorable changes in electronic structure of amide grafted carbon. While the work is of interest to electrocatalysts community, some important details have not been reported while some data is inconsistent and hence the work is not fully convincing:

i) Further details on the computational work should be provided to understand how the alkaline condition (line 131) are constructed.

ii) The authors have used commercial carbon fiber cloth to synthesize the metal-free aminated material. However, commercial carbonaceous materials often contain some metal impurities which can contribute the observed catalytic activity. Therefore, analysis of metal species should be provided for the pristine material and the grafted one obtained after the synthesis in an autoclave.

iii) Electrochemical data provided in Fig. 5a and 5b is contradictory and not fully convincing. The reference material Pt/C performance is greatly affected by number of factors and the data presented in 5a is not in agreement with the state-of-the-art performances reported for Pt/C (see e.g. J. Am. Chem. Soc., 2013, 135 (25), pp 9267 and Angew. Chem. Int. Ed. 2014, 53, 14433). Moreover, the Tafel slope presented in Fig. 5b are contradictory to data plotted in Fig. 5a. The same applies for data presented in Fig. 5d and 5e. Moreover, in the main text (line 206) it is stated that Pt foil has been used as reference but in Fig. 5 20 wt-% Pt/C is given. Which one is the reference? BTW, mentioned inset in Fig. 5c is missing.

iv) In which potential range stability cycling (line 221) is carried out? Electrochemical data supporting stability of the investigated material should be provided. Moreover, XPS data in Fig. 4 for the uncycled material and in SI Fig. 10 for the cycled material differ. In 400 eV zone ammonium peak is lacking for the cycled material. For the pristine material O1s range is not provided and for the cycles C1s so comparison of these is not possible. Hence, the statement about the durability of the material should be better justified.

v) EIS has been carried out at the OCP but the OCP value has not been reported (also the magnitude of the impulse is not reported). In any case, it is implausible that the studied reaction is really proceeding at the OCP. Hence, the measured EIS does not provide information on the HER kinetics as reported by the authors. Furthermore, electrolyte resistance, R_s , should be in the same range for all the investigated materials in case the electrochemical measurement set-up is reproducible.

Reviewer #2 (Remarks to the Author):

In this work, the authors utilized the CFC to grow amide functionalites for the HER studies in alkaline and acidic conditions. Although work looks good but the electrocatalysis part is not up to the mark for high standard journal like this and lot of scientific explanations are not given what they claimed. Therefore, I recommend 'Rejection' in its current form. If author revised the MS as suggested it can be re-submitted to the same journal.

My specified comments are given below.

Comments

1. The authors claimed the formation of OOH group over the carbon from CFC utilizing HNO₃. How it directs the formation of COOH. Is there any possibility of formation of CO₂, NO₂ along with COOH group?
2. The replacement of OH group from COOH was directed by the addition of ammonia. But generally it involves the protonation of ammonium and ending in the formation of NH₄⁺ ions. How the authors claim that OH is replaced by ammonia.
3. There is no information on the loading of the catalyst and the area subjected for the HER study and how the normalization was done.
4. The HER study was demonstrated without any information on iR drop compensation and the % of iR compensation.
5. The mass activity of the catalysts taken has to be analyzed and compared with the previous reports.
6. As an important parameter, Turn Over Frequency (TOF) of the catalysts has to be analyzed.
7. The faradaic efficiency of the catalysts has to be observed to know the activity of the catalysts.
8. The long term stability studies such as Chronoamperometry or chronopotentiometry has to be done to know the real scale stability of the catalyst. But such study was not carried out here.
9. Color mapping for post HER studies in both the electrolytes need to be given to know the percentage of Nitrogen.
10. The electrochemical impedance (EIS) after the cycling study was not provided to know the changes occurred at the interface.
11. Other than theoretical study, there is no much deeper explanation on the electrochemical studies in both the electrolytes.
12. Provide information on the percentage of N, C and O based on XPS analysis for the post HER studies.
13. There are typo errors found throughout the manuscript and rectify it.

Reviewer #3 (Remarks to the Author):

The authors reported an innovative way to transform CFC substrates into amidated-carbon fibers for HER. The theoretical calculations results reveal that surface modification by functional group within this trend provide self-activated electron-extraction character, which is actualized from a fast reversible bond-switching between HO-C=Ccatalyst and O=C-Ccatalyst. This work also demonstrated that high-performance for HER can be achieved by the metal-free electrode. Considering the importance of HER

and the potential for practical applications, the manuscript could be published in Nature Communications after major revisions as noted below:

1. The authors should check the manuscript carefully since there are so many grammatical problems in the main text, for example, Page 1, line 3 and 4; Page 2, line 3 and 4; Page 4, line 2. Except for these grammatical errors, there are many other problems.
2. In abstract part, the authors described 2D, can you explain it?
3. Similarly, in the abstract, introduction, and conclusion, the authors described that the electrocatalyst exhibited high performance under broad-spectrum pH conditions. Is it possible to determine whether the catalyst is active under neutral conditions? If possible, please supplement the results.
4. In the introduction, the authors illustrated that “the high cost and scarcity of the metal-based electrocatalysts severely restrict their large-scale commercial applications”. Please confirm the price of the non-precious metal electrocatalysts is higher than the amidated-carbon fibers. Meanwhile, the authors can provide the price of amidated-carbon fibers.
5. There were many functional groups carbon materials. Can the authors provide the infrared spectroscopy of the CFC, c-CFC, and A-CFC, confirming these groups and changes before and after test?
6. In Fig. 4c, the mapping of N and C did not match well, the authors should supply other mapping pictures.
7. In Page 11, the authors illustrated “The HER performance of A-CFC as a three-dimensional electrode was investigated in H₂-saturated alkaline condition (0.5 M H₂SO₄) using a typical three-electrode system”. 0.5 M H₂SO₄ should be in acid solution.
8. The activity of Pt/C in 1 M KOH was relatively smaller than many reports reported. Please confirm its activity.
9. Please modify “100 KHz to 0.01 Hz”.
10. “The O from the C=O branch will form a weak inter-molecular H-bond to further stabilize the H from the NH₂ branch from nearby group”. The number 2 should be subscript.
11. In table S1, the performance of the other catalysts was all much lower than A-CFC. However, as far as well know, many reports republished in Nature Common. Was much higher than A-CFC. May be the authors can cite papers reported by Nature Common. The papers with high-performance should be cited.
12. Please provide the Faraday efficiencies of A-CFC under acid and alkaline conditions.

Response to Reviewers

Dear Reviewers,

Thanks for spending your precious time to offer the sincere advices and constructive comments on our manuscript entitled "**Rationally engineered active sites for efficient and durable hydrogen generation**". Your advices about the technical details and language corrections of the paper are very helpful for us to improve the quality of our manuscript. Based on your revision suggestions, we have made the following point-by-point response and carefully made a corresponding revision of the manuscript.

The detailed point-by-point responses to all requests are listed as follows:

Reviewer #1 (Remarks to the Author):

The authors suggest amidated-carbon fibers as a metal-free electrocatalyst for hydrogen evolution reaction (HER). Impetus to synthesizing this material is based on the reported computational study showing favorable changes in electronic structure of amide grafted carbon. While the work is of interest to electrocatalysts community, some important details have not been reported while some data is inconsistent and hence the work is not fully convincing:

i) Further details on the computational work should be provided to understand how the alkaline condition (line 131) are constructed.

Response: Thanks for the comments. The answer to the question why HER in alkaline electrolytes is more complicated carries in the requirements of high overpotentials to initiate the catalysis. The additional energy barrier for water dissociation as well as the poor water binding energies is still containing great challenges. Rather than the simple calculations of adsorption of hydrogen, we have calculated the water splitting process which has considered the binding behaviors of water and OH group. The HER in alkaline usually include three types of reactions as Volmer reaction, Heyrovsky reaction and the Tafel reactions that are described as follows [1-2]:

or

where “Catal” denotes to the surface of catalyst and “Catal-H*” depicts hydrogen intermediates adsorbed on the surface. The onset potential of HER is determined by a Volmer step. When the Volmer, Heyrovsky, or Tafel step is the rate-determining step, respectively, different Tafel slopes can be derived from Butler–Volmer kinetics [3-4]. Therefore, by compare the free energy diagram of these reactions on different surfaces, the preferred catalyst with facile barriers can be determined. In this work, we have calculated the kinetic energy barrier of the prior Volmer step [$\Delta G(\text{H}_2\text{O})$] and the concomitant combination of adsorbed H into molecular hydrogen [$(\Delta G(\text{H}))$, Heyrovsky or Tafel step]. The reduced $\Delta G(\text{H}_2\text{O})$ and $\Delta G(\text{OH})$ values on catalysts can suggest that the kinetics of the initial water dissociation step and the concomitant desorption of the formed OH can be effectively promoted, which can turn a sluggish Volmer–Tafel step to a fast Tafel step reaction. **The optimized geometries of our models along the reaction coordinates are following stringent mass transfer with the charge balance in HER of the alkaline condition, which should be distinguished with the simple acidic condition.** Moreover, the calculation of HER in alkaline condition can facilitate the understanding of the bi-functional catalysts that can apply in the pH-universal environment.

Therefore, the simulation of the alkaline environment with the involvement of water and OH in the HER is very critical to precisely reveal the reactivity of catalyst in the reactions.

For all the DFT calculation on Gibbs free energies in this work, all the changes of entropy and zero-point energy have been taken into account. The energy profile is plotted based on:

$$\Delta G = \Delta E - T\Delta S + \Delta ZPE \quad (\text{R1})$$

where ΔE is the reaction energy, ΔS is the entropy change, and ΔZPE is the zero-point energy difference with intrinsic adsorbates-catalyst surface phonon vibration properties considered at the zero-temperature. From the energy diagram, we can clearly observe the binding preference, overbinding and desorption difference in different surfaces. The related formation energies have been computed based on the equation developed by Zunger *et al.* [5] within the CASTEP package. The overall supercell was established and remained fixed for all lattice parameters based on the ground state relaxed primitive cell to reduce the effect of enthalpy changes resulting from cell variations. All the geometry optimizations have been operated

based on the Broyden-Fletcher-Goldfarb-Shannon (BFGS) algorithm. For those non-neutral charge calculations, a Coulomb potential correction will be required to offset the impacts from the image charge of the crystal lattice. The PBE functional was chosen for PBE+U calculations with kinetic cutoff energy of 750 eV, with the valence electron states expressed in a plane-wave basis set. The ensemble DFT (EDFT) method of Marzari *et al.*[6] is used for convergence on the transition metal contained compounds. The RKKJ method is chosen for the optimization of the pseudopotentials [7].

All the detailed computational setup has been provided as Supplementary Materials in the revised manuscript.

References:

- [1] Subbaraman R1, Tripkovic D, Strmcnik D, Chang KC, Uchimura M, Paulikas AP, Stamenkovic V, Markovic NM., *Science* **334**, 1256–1260 (2011).
- [2] Danilovic N, Subbaraman R, Strmcnik D, Chang KC, Paulikas AP, Stamenkovic VR, Markovic NM., *Angew. Chem. Int. Ed.* **124**, 12663–12666 (2012).
- [3] Conway, B. E.; Tilak, B. V., *Electrochim. Acta* **47**, 3571–3594 (2002).
- [4] Mahmood, N.; Yao, Y.; Zhang, J.-W.; Pan, L.; Zhang, X.; Zou, J.-J., *Adv. Sci.* **5**, 1700464 (2017).
- [5] S. Lany; A. Zunger, *Phy. Rev. B*, **78**, 235104 (2008).
- [6] N. Marzari, D. Vanderbilt, M. C. Payne, *Phys. Rev. Lett.* **79**, 1337 (1997).
- [7] A. M. Rappe, K. M. Rabe, E. Kaxiras, J. Joannopoulos, *Phys. Rev. B* **44**, 13175 (1991).

ii) The authors have used commercial carbon fiber cloth to synthesize the metal-free aminated material. However, commercial carbonaceous materials often contain some metal impurities which can contribute the observed catalytic activity. Therefore, analysis of metal species should be provided for the pristine material and the grafted one obtained after the synthesis in an autoclave.

Response: Each carbon fiber in the CFC used in this work consists exclusively of carbon. The atomic structure of carbon fiber is similar to that of graphite, consisting of sheets of carbon atoms arranged in a regular hexagonal pattern (graphene sheets).

As can be seen in Fig. 4d, X-ray photoelectron spectroscopy (XPS) survey of the amide group functionalized CFC (A-CFC) shows the presence of only C, N, and O elements, while the XPS survey of the c-CFC (the precursor) has only C and O elements. The detection limit of the XPS apparatus reaches 0.01%.

In addition, the samples were measured by inductively coupled plasma mass spectrometry (ICP-MS) (Thermo iCAP RQ). The detection limits reach the 0.01 ng g⁻¹ levels for the elements determined.

These results solidly demonstrated that there are no any metal impurities in the materials used in our work.

iii) Electrochemical data provided in Fig. 5a and 5b is contradictory and not fully convincing. The reference material Pt/C performance is greatly affected by number of factors and the data presented in 5a is not in agreement with the state-of-the-art performances reported for Pt/C (see e.g. *J. Am. Chem. Soc.*, 2013, 135 (25), pp 9267 and *Angew. Chem. Int. Ed.* 2014, 53, 14433). Moreover, the Tafel slope presented in Fig. 5b are contradictory to data plotted in Fig. 5a. The same applies for data presented in Fig. 5d and 5e. Moreover, in the main text (line 206) it is stated that Pt foil has been used as reference but in Fig. 5 20 wt-% Pt/C is given. Which one is the reference? BTW, mentioned inset in Fig. 5c is missing.

Response: Thanks for your comments. The commercial Pt/C (20 wt%) was used as reference material for comparison purposes in the HER experiments. According to the recommended references, the polarization curves of Pt/C were re-measured and the Tafel slopes were then determined according to the newly recorded polarization curves. The inset in Fig. 5c was added in Fig. 5 in the revised text.

These have been added in the revised manuscript.

iv) In which potential range stability cycling (line 221) is carried out? Electrochemical data supporting stability of the investigated material should be provided. Moreover, XPS data in Fig. 4 for the uncycled material and in SI Fig. 10 for the cycled material differ. In 400 eV zone ammonium peak is lacking for the cycled material. For the pristine material O1s range is not provided and for the cycles C1s so comparison of these is not possible. hence, the statement about the durability of the material should be better justified.

Response: The stability cycling test (line 221) was carried out between -0.5 V and 0 V versus the saturated calomel electrode (SCE) at 100 mVs⁻¹.

The high resolution O 1s XPS spectra of both c-CFC and A-CFC have already provided as Supplementary Figure (Supplementary Figure 4). Corresponding discussion can be found in the main body of the article.

The XPS data for the cycled material have been reprocessed and analyzed. As shown in Figure R1, the high-resolution N 1s spectrum of the cycled sample showed two peaks at 399.5 and 401.7 eV corresponding to neutral amine nitrogen and ammonium ions, confirming that the nitrogen is still in the amide form even after cycling tests. This was also verified by the O 1s spectra (Figure R2b). C1s peak can also be fitted into five peaks at 284.4, 285.2, 286.9, 288.8 and 290.9 eV, respectively, corresponding to aliphatic carbon chains (C-C,H), amine groups (C-N), carbonyl (O=C), carboxyl carbon (O=C-N), and the π - π^* transitions in amides. The XPS results reveal that the C 1s, N 1s and O 1s have no obvious difference, indicating structural stability of the sample.

More discussions have also been added in the revised manuscript.

Figure R1 | High resolution **a** N 1s, **b** O 1s and **c** C 1s XPS spectra of A-CFC after 18000 cycles in 1.0 M KOH.

v) EIS has been carried out at the OCP but the OCP value has not been reported (also the magnitude of the impulse is not reported). In any case, it is implausible that the studied reaction is really proceeding at the OPC. Hence, the measured EIS does not provide information on the HER kinetics as reported by the authors. Furthermore, electrolyte resistance, R_s , should be in the same range for all the investigated materials in case the electrochemical measurement set-up is reproducible.

Response: According to the Reviewer's suggestion, the EIS data were re-measured at the applied potential of -0.1 V (versus RHE) in the frequency range from 100 kHz to 1 Hz with a signal amplitude perturbation of 5 mV.

The EIS data were fitted by the two-time constant parallel model (2TP, Figure R2 and Table R1) containing the solution resistance (R_s), the charge transfer resistance (R_{ct}), and the

hydrogen adsorption resistance (R_h). The A-CFC shows much smaller R_s (3.14Ω) and R_{ct} (0.69Ω) than that of pure c-CFC ($R_s = 5.39 \Omega$; $R_{ct} = 0.81 \Omega$), suggesting the facilitated HER kinetics. The much smaller R_h of A-CFC reveals the more efficient for the hydrogen intermediates adsorption on the catalyst surface, which is beneficial to the overall HER performances.

Figure R2 | Nyquist curves and fitting-figures of the A-CFC and c-CFC (inset: enlargement of the high frequency region).

Table R1 | Summary of EIS fitting parameters for different samples.

Catalysts	R_s (Ω)	R_{ct} (Ω)	n_1	Q_{ct}	R_h (Ω)	n_2	Q_h
A-CFC	3.14	0.69	0.89	8.7×10^{-4}	245.7	0.95	8.1×10^{-4}
c-CFC	5.39	0.81	1	9.8×10^{-5}	453.7	0.85	1.1×10^{-3}

Reviewer #2 (Remarks to the Author):

In this work, the authors utilized the CFC to grow amide functionalities for the HER studies in alkaline and acidic conditions. Although work looks good but the electrocatalysis part is not up to the mark for high standard journal like this and lot of scientific explanations are not given what they claimed. Therefore, I recommend 'Rejection' in its current form. If author

revised the MS as suggested it can be re-submitted to the same journal. My specified comments are given below.

1. The authors claimed the formation of OOH group over the carbon from CFC utilizing HNO_3 . How it directs the formation of COOH. Is there any possibility of formation of CO_2 , NO_2 along with COOH group?

Response: Thanks for your comments. It is well-known that the carbon materials can be easily oxidized by the concentrated nitric acid. During oxidation process, in principle, carbonyl pairs can be directly formed by breaking the C-C bonds. With further reaction, carboxyl (COOH) groups formed [8–11].

We have paid attention to monitor the reaction. There are no CO_2 and NO_2 formation.

References:

- [8] A. M. Dimiev, and Siegfried Eigler (2017), Graphene Oxide: Fundamentals and Applications. John Wiley & Sons, Ltd..
- [9] Li, Z., Zhang, W., Luo, Y., Yang, J. & Hou, J. G. How graphene is cut upon oxidation? *J. Am. Chem. Soc.* **131**, 6320–6321 (2009).
- [10] Dreyer, D. R., Park, S., Bielawski, C. W. & Ruoff, R. S. The chemistry of graphene oxide. *Chem. Soc. Rev.* **39**, 228–240 (2010).
- [11] Jankovský, O. *et al.* Concentration of nitric acid strongly influences chemical composition of graphite oxide. *Chem. Eur. J.* **23**, 6432–6440 (2017).

2. The replacement of OH group from COOH was directed by the addition of ammonia. But generally it involves the protonation of ammonium and ending in the formation of NH_4^+ ions. How the authors claim that OH is replaced by ammonia?

Response: The direct reaction of a carboxylic acid with an amine would be expected to be difficult because the basic amine would deprotonate the carboxylic acid to form a highly unreactive carboxylate. However, **when the ammonium carboxylate salt is heated to a temperature above 100 °C, water is driven off and an amide is formed** [12,13].

As detailedly described in the “Methods” section: The freshly-prepared c-CFC was immediately into another Autoclave (50 mL) containing 40 mL ammonium hydroxide solution (28% NH_3 in H_2O). After 20 minutes' standing, the sealed Autoclave was placed **at**

150 °C for 10 h, then was naturally cooled to room temperature and washed thoroughly with Millipore water, followed by being dried in a vacuum oven for 2 hours. The A-CFC was obtained.

Scheme R1 | A general route to the conversion of a carboxylic acid to an amide.

References:

[12] Qu, D. *et al.* Formation mechanism and optimization of highly luminescent N-doped graphene quantum dots. *Sci. Rep.* **4**, 5294 (2014).

[13] Rushdi, A. I. & Simoneit, B. R. Condensation reactions and formation of amides, esters, and nitriles under hydrothermal conditions. *Astrobiology.* **4**, 211–224 (2004).

3. There is no information on the loading of the catalyst and the area subjected for the HER study and how the normalization was done.

Response: Thanks for your reminder. The average mass loading of A-CFC is 1.25 mg cm⁻². The A-CFC with 0.1 cm⁻² (geometric area) was used as the working electrode and the HER performances were investigated using a typical three-electrode system.

For comparison, c-CFC with the similar mass loading (1.30 mg cm⁻²) and the same geometric (0.1 cm⁻²) was used the cathodic electrode. Current density normalized to geometric area of the working electrode.

These has been provided in the revised manuscript.

4. The HER study was demonstrated without any information on iR drop compensation and the % of iR compensation.

Response: All LSV curves with 100% iR compensation were obtained. Without further specified, all potentials were relative to the reversible hydrogen electrode (vs. RHE): E_{RHE} = E_{measured} - iR + E_{SCE} + 0.059*pH, where E_{measured} is the measured potential, i is the current, and R is the ohmic drop tested by electrochemical impedance spectroscopy (EIS).

The information was added to the revised version.

5. The mass activity of the catalysts taken has to be analyzed and compared with the previous reports.

Response: The mass activity of a catalyst is calculated by the following equation:

$$J_{mass} = I/M$$

where I is the current we get directly from the electrochemical tests at the corresponding voltage for HER. M is the loaded mass of catalyst per geometrical area of electrode.

It should be mentioned that, for our reported catalyst, the active sites have been demonstrated to be related to the amide groups over the whole surface of the working electrode. The amide group provides a dynamic self-activated electron-extraction character to facilitate the 2D electronic distribution from the graphene towards to be more localized on the group, especially at the C=O branch.

Therefore, a rough calculation on the mass activity of A-CFC was carried out based on the average mass loading (1.25 mg cm^{-2}). Figure R3 shows the mass activities of the catalysts. For example, at the overpotential of 150 mV, the mass activities of A-CFC are 186.1 mA mg^{-1} and 43.2 mA mg^{-1} in $0.5 \text{ M H}_2\text{SO}_4$ and 1 M KOH , respectively. These values are higher than most of the recently reported electrocatalysts,

Figure R3 | Mass activities of the catalysts obtained in **a** $0.5 \text{ M H}_2\text{SO}_4$ and **b** 1 M KOH .

Table R2 | Comparison of selected state-of-the-art metal-free HER electrocatalysts in acidic/alkaline aqueous media.

Catalysts	Mass activities	Electrolytes	References
-----------	-----------------	--------------	------------

A-CFC	186.1 mA mg ⁻¹ at 150 mV 353.6 mA mg ⁻¹ at 200 mV	0.5 M H ₂ SO ₄	This work
	43.2 mA mg ⁻¹ at 150 mV 119 mA mg ⁻¹ at 300 mV	1 M KOH	
C ₃ N ₄ @NG	100 mA mg ⁻¹ at 240 mV	0.5 M H ₂ SO ₄	Nat. Commun. 4 , 3783 (2014)
NS-doped hierarchical nanoporous graphene	28.6 mA mg ⁻¹ at 230 mV	0.5 M H ₂ SO ₄	Angew. Chem. Int. Ed. 57 , 13302–13307 (2018)
B/P/S-Doped g-C ₃ N ₄	35.7 mA mg ⁻¹ at 186 mV	0.5 M H ₂ SO ₄	ACS Nano 11 , 6004–6014 (2017)
g-CN@G MMs	70.4 mA mg ⁻¹ at 219 mV	0.5 M H ₂ SO ₄	Adv. Funct. Mater. 27 , 1606352 (2017)
Defective graphene	35.1 mA mg ⁻¹ at 320 mV	1 M KOH	Adv. Mater. 28 , 9532–9538 (2016)
N, P and O tri-doped porous graphite carbon@oxidized carbon cloth	100 mA mg ⁻¹ at 450 mV	1 M KOH	Energy Environ. Sci. 9 , 1210–1214 (2016)

6. As an important parameter, Turn Over Frequency (TOF) of the catalysts has to be analyzed.

Response: Thank you for your advice. The TOF is calculated according to the following equation:

$$\text{TOF} = \frac{\text{Total number of H}_2 \text{ molecules per second}}{\text{Total number of active sites per unit area}} = \frac{j}{2qN}$$

where j is the current density, q is the elementary charge as 1.6×10^{-19} , N is the active site density, and 2 means the electron transfer number in one hydrogen molecule generation. The active sites per unit area can be estimated from the electrochemical surface area (ECSA = 235 cm², please see Methods Section in the manuscript for details).

According to our experimental results, the active sites are related to the amide groups. The upper limit number of HER active sites in A-CFC was calculated based on the hypothesis that N atoms on the A-CFC surface form the active centers, and are all accessible to the electrolyte. The percentage of N was obtained from the EDS results (~1.44 wt%). The average mass of

the A-CFC electrode is 0.125 mg. The active sites density (N) was calculated to be 7.74×10^{17} sites cm^{-2} .

Therefore, the TOF for our catalyst at different overpotentials is calculated as follows:

In 0.5 M H_2SO_4 ,

$$\text{At 100 mV, TOF} = \frac{j}{2qN} = \frac{j}{2 \times 1.6 \times 10^{-19} \times 7.74 \times 10^{17} \times 235} = \frac{50.1}{58.2} = 0.86 \text{ s}^{-1}$$

In 1 M KOH,

$$\text{At 100 mV, TOF} = \frac{24.11}{58.2} = 0.41 \text{ s}^{-1}$$

It can be observed that the TOF of the A-CFC catalyst is 0.86 and 0.41 s^{-1} at the overpotential of 100 mV in 0.5 M H_2SO_4 and 1.0 M KOH, respectively, which are more than 10-fold and 2-fold larger than the c-CFC (0.08 s^{-1} in 0.5 M H_2SO_4 and 0.14 s^{-1} in 1.0 M KOH). This significant increase of TOF reveals an improved energetics at the amidated interface during the hydrogen-evolving process, leading to the enhancement of HER catalytic performance of the A-CFC sample.

These additional information and discussion have been supplied in the revised manuscript.

7. The faradaic efficiency of the catalysts has to be observed to know the activity of the catalysts.

Response: The amount of hydrogen gas evolved in the electrolysis cell was measured with the Faraday efficiencies over 97% in both 0.5 M H_2SO_4 and 1 M KOH electrolytes (Figure R4), respectively. These have been added in the revised Manuscript.

Figure R4 | The evolved H₂ measured with GC (red) and theoretical volume (black) versus time obtained in **a** 0.5 M H₂SO₄ and **b** 1 M KOH condition. Faraday efficiencies higher than 97 % (calculate on the data of 6000 s).

8. The long term stability studies such as Chronoamperometry or chronopotentiometry has to be done to know the real scale stability of the catalyst. But such study was not carried out here.

Response: The long term stability studies of the catalyst were conducted in 0.5 M H₂SO₄ and 1 M KOH, respectively. As shown in Figure R5, in 0.5 M H₂SO₄, the current density decreased at the beginning and then stabilized over 28 h; while the A-CFC remains stable over the whole electrolysis process in 1 M KOH.

Figure R5 | Chronopotentiometric curves of A-CFC recorded in **a** 0.5 M H₂SO₄ and **b** 1 M KOH for HER.

9. Color mapping for post HER studies in both the electrolytes need to be given to know the percentage of Nitrogen.

Response: Color mapping for A-CFC obtained after HER studies were performed. The elemental mapping results for samples obtained after HER studies indicated the C, N, O elements were the predominant elements in respective sample. The percentages of Nitrogen obtained after HER studies in 1 M KOH (Figure R6) and 0.5 M H₂SO₄ (Figure R7) are 1.44 wt.% and 1.18 wt.%, respectively.

These results have been provided as Supplementary Figures in the revised version.

Figure R6 | **a** Scanning TEM image, **b-e** elemental mapping, and **f** the percentage of elemental composition in A-CFC obtained after HER test in 1 M KOH.

Figure R7 | **a** Scanning TEM image, **b-e** elemental mapping, and **f** the percentage of elemental composition in A-CFC obtained after HER test in 0.5 M H₂SO₄.

10. The electrochemical impedance (EIS) after the cycling study was not provided to know the changes occurred at the interface.

Response: Thanks for your suggestion. The electrochemical impedance spectroscopy (EIS) after the cycling test was measured (Figure R8) and fitted by the two-time constant parallel model (2TP). The fitting parameters are illustrated in Table R3. Compared with the freshly-prepared A-CFC, there are slight increase in both R_s and R_{ct} after cycling study.

Table R3 | Summary of EIS fitting parameters for A-CFC before and after cycling test.

Catalysts	R_s (Ω)	R_{ct} (Ω)	n_1	Q_{ct}	R_h (Ω)	n_2	Q_h
A-CFC (before cycling test)	3.14	0.69	0.89	8.7×10^{-4}	245.7	0.95	8.1×10^{-4}
A-CFC (after cycling test)	3.24	0.83	1	5.5×10^{-4}	132	0.90	1.0×10^{-3}

Figure R8 | Nyquist curves of the A-CFC before and after cycling test (inset: enlargement of the high frequency region).

11. Other than theoretical study, there is no much deeper explanation on the electrochemical studies in both the electrolytes.

Response: Thanks for your suggestion. More detailed discussions on the electrochemical studies in both the electrolytes have been added in the revised manuscript.

12. Provide information on the percentage of N, C and O based on XPS analysis for the post HER studies.

Response: Table R4 displays the percentage of N, C and O based on XPS analysis for A-CFC after HER studies in 0.5 M H₂SO₄ and 1 M KOH.

Table R4 | The percentage of N, C and O based on XPS analysis for A-CFC after HER studies in 0.5 M H₂SO₄ and 1 M KOH.

Electrolyte	C (atomic %)	N (atomic %)	O (atomic %)
0.5 M H ₂ SO ₄	95.13	1.33	3.54
1 M KOH	91.53	1.25	7.22

13. There are typo errors found throughout the manuscript and rectify it.

Response: Thanks for your kind reminder. The manuscript has been carefully checked and typo errors have been corrected.

Reviewer 3:

The authors reported an innovative way to transform CFC substrates into amidated-carbon fibers for HER. The theoretical calculations results reveal that surface modification by functional group within this trend provide self-activated electron-extraction character, which is actualized from a fast reversible bond-switching between HO-C=Ccatalyst and O=C-Ccatalyst. This work also demonstrated that high-performance for HER can be achieved by the metal-free electrode. Considering the importance of HER and the potential for practical applications, the manuscript could be published in Nature Communications after major revisions as noted below:

1. The authors should check the manuscript carefully since there are so many grammatical problems in the main text, for example, Page 1, line 3 and 4; Page 2, line 3 and 4; Page 4, line
2. Except for these grammatical errors, there are many other problems.

Response: Thanks for your useful suggestions on the language. The problems have been corrected in the revised manuscript. We have revised the WHOLE manuscript carefully and update some English proof reading with help of native speaker (Dr.Alan William Dougherty) from Dr. Bolong Huang's group. We believe that the language is now acceptable for the

review process.

2. In abstract part, the authors described 2D, can you explain it?

Response: 2D electrons in this work can be categorized as the two-dimensional delocalized π -like electron gas on graphene, which will extract the electronic distribution from the surface towards the surface groups, especially electron-rich characterized C=O branch. With the surface modification, the long range ordered p-p resonance of graphene will be weakened or interrupted, which can activate the redistribution of these delocalized electrons towards the adsorbates. Thus, the reversible bond-switching of delocalized 2D electrons induced by surface modifications are the pivotal to boost HER.

3. Similarly, in the abstract, introduction, and conclusion, the authors described that the electrocatalyst exhibited high performance under broad-spectrum pH conditions. Is it possible to determine whether the catalyst is active under neutral conditions? If possible, please supplement the results.

Response: The catalytic performance of the electrocatalysts have been determined in neutral conditions (1 M PBS, pH 7), as shown in Figure R9. A-CFC exhibits small overpotentials of 230.8 and 436 mV at the current densities of 2 and 10 mA cm⁻², respectively, and Tafel slope of 263.8 mV dec⁻¹. These values are better than that of c-CFC (570 and 791 mV at 2 and 4 mA cm⁻², respectively; Tafel slope: 730.4 mV dec⁻¹), and superior to that of non-metal and transition metal based electrocatalysts (Table R5).

Corresponding discussions have been added in the revised version.

Figure R9 | **a** HER polarization curves and **b** corresponding Tafel plots of catalysts obtained

in 1 M PBS.

Table R5 | Comparison of the HER activity of the A-CFC in 1 M PBS with other reported metal-free and transition metal-based electrocatalysts.

Electrocatalysts	j (mA cm ⁻²)	η (mV) @ j	References
A-CFC	2	230.8	This work
	10	436	
PPANI750	10	~580	J. Am. Chem. Soc. 137 , 15070-15073 (2015).
Co ₃ S ₄	10	~480	J. Am. Chem. Soc. 138 , 1359-1365 (2016)
H ₂ -CoCat film	2	385	Nat. Mater. 11 , 802-807 (2012).
Co-S/FTO	10	~720	J. Am. Chem. Soc. 135 , 17699-17702 (2013)
Co-NRCNTs	10	~ 540	Angew. Chem. Int. Ed. 53 , 4372-4376 (2014)

4. In the introduction, the authors illustrated that “the high cost and scarcity of the metal-based electrocatalysts severely restrict their large-scale commercial applications”. Please confirm the price of the non-precious metal electrocatalysts is higher than the amidated-carbon fibers. Meanwhile, the authors can provide the price of amidated-carbon fibers.

Response: Thank you for the comments. Due to the different fabrication method and processing cost in most of the works, it is very difficult to directly compare the price between both non-precious metal electrocatalysts and the amidated-carbon fibers. However, we have obtained direct prices of some typical non-precious metals and carbon fibers in the past one year (from Feb 1, 2018, to Jan 14, 2019) for the reference comparison purpose, which are summarized and illustrated in Table R6. **Note:** Data are obtained from publicly available websites (<http://www.infomine.com>; <https://www.alibaba.com>), for reference only.

It's found that the prices of some non-precious metals are lower than the amidated-carbon fibers. But, the prices of all precious metals are higher than the amidated-carbon fibers.

So, the sentence “the high cost and scarcity of the metal-based electrocatalysts severely restrict...” was rewritten as follows: “the high cost and scarcity of the noble-metal-based electrocatalysts severely restrict their cosmically large-scale commercial applications.”.

Table R6 | The prices of typical non-precious metals and carbon fibers.

Materials		Price	
Non-precious metals	Copper	6,352.62	USD/t
	Nickel	12,350.08	USD/t
	Cobalt	31,000.08	USD/t
	Molybdenum	26,000.00	USD/t
Precious metals	Gold	1,344.00	USD/ozt
	Palladium	1,497.40	USD/ozt
	Platinum	820.50	USD/ozt
	Ruthenium	266.00	USD/ozt
	Iridium	1,460.00	USD/ozt
	Rhodium	2,645.00	USD/ozt
Carbon fibers	Carbon fiber	13,800.00	USD/t

5. There were many functional groups carbon materials. Can the authors provide the infrared spectroscopy of the CFC, c-CFC, and A-CFC, confirming these groups and changes before and after test?

Response: Thanks for your suggestion. The Fourier transform infrared (FT-IR) spectroscopy was used to track the variation of the functional groups. Figure R10 shows the FT-IR spectra of CFC (Figure R10a), c-CFC (Figure R10b), and A-CFC (Figure R10a). For the CFC, the characteristic peak located at around 3450 cm^{-1} assigned to the O-H stretching mode of the absorbed water. After the oxidation was executed, the peak located at 1721 cm^{-1} was the characteristic of -COOH (C=O in carboxylic acid), revealing the successful achievement of c-CFC. The peak at $\sim 2971\text{ cm}^{-1}$ could be assigned to to the O-H stretching mode of -COOH

groups. As shown in Figure R10c, the peaks located at 3388 and 1619 cm^{-1} , correspond to the stretching and bending modes of the N-H.

Figure R10 | FT-IR spectra of **a** CFC, **b** c-CFC, and **c** A-CFC.

6. In Fig. 4c, the mapping of N and C did not match well, the authors should supply other mapping pictures.

Response: Thanks for your kind advice. As shown in Figure R11, new elemental mapping tests were conducted. New mapping pictures have been supplied in the revised manuscript.

Figure R11 | Scanning TEM image and corresponding elemental mapping of N (yellow) and C (green) elements in A-CFC.

7. In Page 11, the authors illustrated “The HER performance of A-CFC as a three-dimensional electrode was investigated in H₂-saturated alkaline condition (0.5 M H₂SO₄) using a typical three-electrode system”. 0.5 M H₂SO₄ should be in acid solution.

Response: This has been corrected in the revised text.

8. The activity of Pt/C in 1 M KOH was relatively smaller than many reports reported. Please confirm its activity.

Response: The polarization curves of Pt/C were re-measured and the Tafel slopes were determined according to the newly recorded polarization curves. These have been updated in the revised text.

9. Please modify “100 KHz to 0.01 Hz”.

Response: This has been modified as “The EIS data were obtained in the same configuration at the applied potential of -0.1 V (versus RHE) in the frequency range from 100 kHz to 1 Hz with a signal amplitude perturbation of 5 mV.”.

10. “The O from the C=O branch will form a weak inter-molecular H-bond to further stabilize

the H from the NH₂ branch from nearby group”. The number 2 should be subscript.

Response: The number 2 has been subscripted.

11. In table S1, the performance of the other catalysts was all much lower than A-CFC. However, as far as well know, many reports republished in Nature Common. was much higher than A-CFC. May be the authors can cite papers reported by Nature Common. The papers with high-performance should be cited.

Response: The A-CFC reported in our work is a metal-free electrocatalyst toward hydrogen evolution reaction (HER). Therefore, the catalysts illustrated in table S1 are almost metal-free electrocatalysts with excellent HER activities. In order to highlight the catalytic activity of A-CFC, some benchmarked metal-based electrocatalysts were employed for comparison. In the revised manuscript, the papers reported by Nature Common. were cited.

12. Please provide the Faraday efficiencies of A-CFC under acid and alkaline conditions.

Response: As shown Figure R4, the amount of hydrogen gas evolved in the electrolysis cell was measured with the Faraday efficiencies over 97% in both 0.5 M H₂SO₄ and 1 M KOH electrolytes (Figure R4), respectively. These have been added in the revised Manuscript.

Thank you again for your consideration and appreciation on our manuscript. It is our honor that you offer so many valuable comments for us to improve the quality of our manuscript for wider range of readers. We really appreciate all the suggestions and advices because they are not only useful to our manuscript but also to our future research. We would be grateful and treasure this opportunity if you can approve our revised manuscript.

Reviewers' comments:

Reviewer #2 (Remarks to the Author):

Title: Rationally engineered active sites for efficient and durable hydrogen generation

Authors: Yurui Xue, Lan Hui, Huidi Yu, Yuxin Liu, Yan Fang, Bolong Huang, Yingjie Zhao, Zhibo Li and Zhuo Chen

As requested by Editor, I checked the Q-A part of Reviewer 1 mainly. For me it looks convincing. Author answer all the Qs with proper experimental proofs and justification. I felt MS is suitable for publication.

Title: Rationally engineered active sites for efficient and durable hydrogen generation

Authors: Yurui Xue, Lan Hui, Huidi Yu, Yuxin Liu, Yan Fang, Bolong Huang, Yingjie Zhao, Zhibo Li and Zhuo Chen

As per the queries rose and the observed proper scientific replies with experimental evidences from the authors, the manuscript can be accepted. However, still there are few more queries to be answered for a wide readership. My specified comments are given below.

Comments

1. The authors did not compare the response in LSV's of pristine CFC with those of A-CFC and C-CFC. Try to include the response of CFC also.
2. What is the nature of the carbon fiber cloth used?
3. In calculating TOF, the information on calculating the active site density (N) is not clear. Give in depth information on the calculation of active site density from the atomic percentage and mass loading studies.
4. The loading of a catalyst at a geometric surface area of 0.1 cm² is 1.25 mg cm⁻². This is too higher than the normalized loading and was there any leaching of catalyst? Authors need to justify this.

Reviewer #3 (Remarks to the Author):

1) In EIS, the R_s is 5.39 ohm. This is too big for iR correction. When the current density is 10 mA/cm², the potential will be corrected 53.9 mV! For HER, this is not acceptable. Such a big potential correction would misleading the readers. A-CFC only has the overpotential of 10 mA/cm² of 71 mV. R_s should be basically not bigger than 1 ohm, otherwise the overpotential for 10 mA/cm² will be adjusted by iR correction! Please double check the HER test. R_s is small, because the solution is conductive. Nature communications published many papers, and they always have a small R_s .

2) The carbon materials are easy oxidized by acid or basic solution (there are many documents). It is not understandable that the authors claim there are changes after stability test of CFC.

Response to reviewers

Dear Reviewers,

Thank you for the sincere advice and comments on our manuscript entitled "**Rationally engineered active sites for efficient and durable hydrogen generation**" for *Nature Communications*. We are very grateful for your valuable suggestions and comments on the technical details, formatting, structure of the paper that are very helpful for us to improve our work. Based on your revision suggestions, we have made the revision on our manuscript and the point-by-point comments are listed below.

The detailed point-by-point responses to all requests are listed as follows:

Reviewer #2 (Remarks to the Author):

Title: Rationally engineered active sites for efficient and durable hydrogen generation

Authors: Yurui Xue, Lan Hui, Huidi Yu, Yuxin Liu, Yan Fang, Bolong Huang, Yingjie Zhao, Zhibo Li and Zhuo Chen

Response: Thank you for the assistance in all the revision and peer review. According to our original submitted manuscript, there is no person named “Zhuo Chen” as co-author in our manuscript. The author named “Zhuo Chen” as a co-author should be removed for the correction.

As requested by Editor, I checked the Q-A part of Reviewer 1 mainly. For me it looks convincing. Author answer all the Qs with proper experimental proofs and justification. I felt MS is suitable for publication.

Response: Thank you very much for your assistants in all the revision and peer review.

Title: Rationally engineered active sites for efficient and durable hydrogen generation

Authors: Yurui Xue, Lan Hui, Huidi Yu, Yuxin Liu, Yan Fang, Bolong Huang, Yingjie Zhao, Zhibo Li and Zhuo Chen

Response: According to our original submitted manuscript, there is no person named “Zhuo Chen” as co-author in our manuscript. The author named “Zhuo Chen” as a co-author should be removed for the correction.

As per the queries rose and the observed proper scientific replies with experimental evidences from the authors, the manuscript can be accepted. However, still there are few more queries to be answered for a wide readership. My specified comments are given below.

Comments:

1. The authors did not compare the response in LSV's of pristine CFC with those of A-CFC and C-CFC. Try to include the response of CFC also.

Response: Thank you very much for your constructive advice. The LSV curves of pristine CFC was measured in both 0.5 M H₂SO₄ and 1.0 M KOH. As shown in Figure R1, pristine CFC requires overpotentials of 577 mV and 611 mV in 0.5 M H₂SO₄ and 1.0 M KOH, respectively, to achieve 10 mA cm⁻², which are much larger than those of A-CFC and c-CFC. These results have been added in the revised manuscript.

Figure R1. The polarization curves of the catalysts obtained in **a** 0.5 M H₂SO₄ and **b** 1 M KOH.

2. What is the nature of the carbon fiber cloth used?

Response: Thank you for the thoughtful question. To illustrate clearly the nature of the carbon fiber cloth used in this work, we have supplied a detailed table (**Table R1**) below with all the basic properties. The information was obtained from the CeTech Co., Ltd. that supplied the carbon fiber for this work.

Table R1. The nature of the carbon fiber cloth.

Measurements	Units	Method	Results
Thickness	mm	TECLOCK SM-114	0.33

Air Permeability	Sec	Gurley	<10
Through-Plane Resistance	$m\Omega cm^2$	Base on ASTM C-611	<5
Tensile Strength (MD)	N/cm	ASTM D-828	10
Tensile Strength (XD)	N/cm	ASTM D-828	5

3. In calculating TOF, the information on calculating the active site density (N) is not clear. Give in depth information on the calculation of active site density from the atomic percentage and mass loading studies.

Response: Thank you for your careful review of our manuscript and suggestions to improve the quality of the manuscript. According to the experimental and theoretical results, the active sites are related to the amide groups. The upper limit number of HER active sites in A-CFC was calculated based on the hypothesis that N atoms on the A-CFC surface formed the active centers accessible to the electrolyte. The percentage of N was obtained from the EDS results (~1.44 wt.%). The average mass of the A-CFC electrode is 0.125 mg, and the geometric surface area of the electrode is 0.1 cm^2 . The active sites density (N) can be calculated according to the following equation:

$$N = \frac{n \times N_A}{A}$$

where n is the mole of the active centers, N_A is the Avogadro constant, and A corresponds to the geometric surface area of the electrode.

Therefore,

$$\begin{aligned} \text{Active sites density (N)} &= \frac{n \times N_A}{A} = \frac{(0.125 \text{ mg} \times 1.44 \text{ wt. \%})/1000}{14 \text{ g mol}^{-1}} \times (6.022 \times 10^{23}) \\ &= 7.74 \times 10^{17} \text{ sites cm}^{-2} \end{aligned}$$

To ensure the accuracy, all these calculation and results have been carefully checked and updated in the revised manuscript.

4. The loading of a catalyst at a geometric surface area of 0.1 cm^2 is 1.25 $mg cm^{-2}$. This is too higher than the normalized loading and was there any leaching of catalyst? Authors need to justify this.

Response: Thanks for the careful review and the concerns on the technical details of this work. The catalyst (A-CFC) is a monolithic electrode constructed through in situ surface modification strategy, in which the amino groups were covalently linked onto the carbon fiber surface, in contrast to drop-casted catalysts, which suffer seriously from the low mass loading and the leaching of catalyst during the test. Besides, after careful comparison of the catalysts before and after the electrochemical measurements, we have confirmed that no leaching of the catalyst has been discovered.

Reviewer #3

1) In EIS, the R_s is 5.39 ohm. This is too big for iR correction. When the current density is 10 mA/cm², the potential will be corrected 53.9 mV! For HER, this is not acceptable. Such a big potential correction would misleading the readers. A-CFC only has the overpotential of 10 mA/cm² of 71 mV. R_s should be basically not bigger than 1 ohm, otherwise the overpotential for 10 mA/cm² will be adjusted by iR correction! Please double check the HER test. R_s is small, because the solution is conductive. Nature communications published many papers, and they always have a small R_s .

Response: Thanks for the careful review and questions on the technical details. To clearly answer your concerns, the following explanation is supplied for your confusion.

First of all, I would like to point out that the **R_s (5.39 ohm) mentioned by the reviewer actually correspond to the c-CFC sample** (not the A-CFC sample), which exhibits an overpotential of 530 mV at 10 mA cm⁻².

Secondly, for the iR correction, we believed that the concerns arose from some technical confusion. In this work, i represents the measured **current** during all measurements rather than the **current density**. Therefore, we believed that our results are reliable to address the reviewer's worries.

R_s is the resistance between the working electrode and the reference electrode. In a typical three-electrode electrochemical test cell, the value of the R_s is affected by many factors including the surface structure of the electrocatalysts, the surface of the counter electrode, the electrolyte surface of the working electrode, electrolyte surface of the reference electrode, etc.

Thus, the value of R_s cannot be simply correlated to one specific reason in the work. Moreover, our literature review suggested that recently reported works in *Nature Communications* also exhibited relatively R_s . Please kindly see in *Nat. Commun.* **7**, 11857 (2016), *Nat. Commun.* **9**, 2120 (2018), *Nat. Commun.* **10**, 631 (2019), and *Nat. Commun.* **10**, 982 (2019).

2) The carbon materials are easy oxidized by acid or basic solution (there are many documents). It is not understandable that the authors claim there are changes after stability test of CFC.

Response: Thank you for the concerns.

We believed that our present experimental results are fully supportive of our conclusion. From the XPS results (Supplementary **Fig. S9** and **S11** in the manuscript), it is clearly revealed that there are no notable chemical changes after the stability tests in both 0.5 M H_2SO_4 and 1.0 M KOH conditions. Thus, these results are sufficient for us to obtain the conclusion of the high stability of CFC. The detailed experiment information is supplied in the manuscript for your concerns on the stability tests.

Thank you again for your consideration and appreciation on our manuscript. We really appreciate all the suggestions and advices because they are not only useful to our manuscript but also to our future research.

REVIEWERS' COMMENTS:

Reviewer #2 (Remarks to the Author):

MS has been revised as suggested. It can be accepted as it is now.

Reviewer #3 (Remarks to the Author):

The authors basically addressed the reviewer's comments, and the paper should be accepted.

Response to reviewers

Dear Reviewers,

Thank you for the sincere advice and comments on our manuscript entitled "**Rationally engineered active sites for efficient and durable hydrogen generation**" for *Nature Communications*. We are very grateful for your valuable suggestions and comments on the technical details, formatting, structure of the paper that are very helpful for us to improve our work.

The detailed point-by-point responses to all requests are listed as follows:

REVIEWERS' COMMENTS:

Reviewer #2 (Remarks to the Author):

MS has been revised as suggested. It can be accepted as it is now.

Response: Thank you very much for your precious time and kind suggestions during the revision and peer review.

Reviewer #3 (Remarks to the Author):

The authors basically addressed the reviewer's comments, and the paper should be accepted.

Response: Thank you very much for the precious time and helpful comments during all the peer review.

Thank you again for your consideration and appreciation on our manuscript. We really appreciate all the suggestions and advices because they are not only useful to our manuscript but also to our future research.